# Quantitative Phosphoproteomics of *cipk3*/*9*/*23*/*26* Mutant and Wild Type in *Arabidopsis thaliana*

**DOI:** 10.3390/genes12111759

**Published:** 2021-11-04

**Authors:** Ziyi Yin, Jisen Shi, Yan Zhen

**Affiliations:** 1Key Laboratory of Forest Genetics and Biotechnology Ministry of Education, Nanjing Forestry University, Nanjing 210037, China; elinyin1224@163.com (Z.Y.); jshi@njfu.edu.cn (J.S.); 2Co-Innovation Center for Sustainable Forestry in Southern China, Nanjing Forestry University, Nanjing 210037, China

**Keywords:** *Arabidopsis thaliana* L., CBL-interacting protein kinases 6, signal transduction, magnesium sensitivity, tandem mass tag labeling

## Abstract

CBL-interacting protein kinases 3/9/23/26 (CIPK3/9/23/26) are central regulation components of magnesium ion homeostasis. CBL2/3 interacts with CIPK3/9/23/26, which phosphorylates their downstream targets, suggesting that protein phosphorylation is a key factor influencing the maintenance of cellular magnesium homeostasis in higher plants. The *cipk3*/*9*/*23*/*26* quadruple mutant is very sensitive to high levels of magnesium. In this study, TMT quantitative phosphoproteomics were used to compare the global variations in phosphoproteins in wild type and *cipk3*/*9*/*23*/*26* quadruple mutant seedlings of *Arabidopsis thaliana*, and 12,506 phosphorylation modification sites on 4537 proteins were identified, of which 773 phosphorylated proteins exhibited significant variations at the phosphorylation level under magnesium sensitivity. Subsequently, we used bioinformatics methods to systematically annotate and analyze the data. Certain transporters and signaling components that could be associated with magnesium sensitivity, such as ATP-binding cassette transporters and mitogen-activated protein kinases, were identified. The results of this study further our understanding of the molecular mechanisms of CIPK3/9/23/26 in mediating magnesium homeostasis.

## 1. Introduction

Plants require all kinds of mineral elements, which are absorbed from soil in the form of ions by the plant roots and transported to various plant tissues through a series of transportation vehicles, to maintain their growth and development [1]. Plants sometimes absorb more or less mineral nutrients from the soil due to the influence of external environmental factors, which forces plants to evolve mechanisms of ion homeostasis that can maintain the dynamic balance of each nutrient ion.

Calcineurin B-like protein (CBL) is an uncommon class of calcium ion (Ca^2+^) sensors in plants, which can physically and functionally interact with the CBL-interacting protein kinase (CIPK) family, and decode various dynamic Ca^2+^ signals and regulate relevant physiological processes by phosphorylating downstream target proteins [2]. Numerous CBL and CIPK family members constitute the CBL-CIPK signaling network, which can facilitate the regulation of ion homeostasis. Studies have revealed that the CBL-CIPK signaling network is a vital factor influencing various physiological activities and multiple signaling pathways in plants, such as salt stress responses, low potassium stress response, regulation of the abscisic acid (ABA) signal transduction, and regulation of plasma membrane H^+^-ATPase activity, among others. For instance, Pandey et al. [3] reported that the growth of the CIPK21 loss-of-function mutant, *cipk21*, exhibited hypersensitivity to high salinity and osmotic stress. The CBL2 and CBL3 Ca^2+^ sensors have been demonstrated to physically interact with CIPK21 and target the kinase to the vacuolar membrane. Studies have revealed that CIPK21 is involved in responses to high salinity in *Arabidopsis thaliana* by balancing ion and water homeostasis on the vacuolar membrane. Sanyal et al. [4] reported that CBL9 could be phosphorylated by CIPK3, and ABA repressor 1 (ABR1) was the downstream target of CIPK3. CIPK3 and ABR1 not only regulate the function of the ABA response during seed germination, but are also involved in ABA-dependent signaling in the adult plant development.

Among the essential nutrient ions, magnesium ion (Mg^2+^) is one of the most abundant metal ions in cells and is a key element necessary for plant growth and development [5]. However, the presence of excess Mg^2+^ in the environment can influence the normal growth and development of plants, causing plants to exhibit severe growth retardation. *cipk3*/*9*/*23*/*26* is a quadruple mutant produced through the hybridization of two double mutants, *cipk9*/*23* and *cipk3*/*26*, which is sensitive to Mg under moderate or high external Mg^2+^ levels. *Arabidopsis* is defective in the four tonoplast-localized cipk3/9/23/26 proteins intolerant to medium and high concentrations of Mg^2+^, which exhibits similar severe growth retardation to that of cbl2 cbl3. Tang et al. [6] revealed a regulatory mechanism, which demonstrated that CBL2/3 and CIPK3/9/23/26 constituted a network of multivalent interactions that enabled plant cells to sequester excess Mg^2+^ into vacuoles, consequently shielding plant cells from high Mg^2+^ toxicity. Mogami et al. [7] revealed that ABA-activated SRK2D/E/I and CIPK26/3/9/23 were essential in *Arabidopsis* for plant growth at high external Mg^2+^ concentrations, and that they mediated phosphorylation signaling pathways that maintained cellular Mg^2+^ homeostasis.

In this study, TMT label-based quantitative phosphoproteomics were used to analyze phosphorylated proteins, with the aim of qualitatively and quantitatively analyzing the variations in phosphoproteomics of wild type (WT) and *cipk3*/*9*/*23*/*26* quadruple mutant seedlings of *Arabidopsis*. The results of the present study provide comprehensive insights into how CIPK3/9/23/26 mediates Mg^2+^ homeostasis and adapts to variations in cellular physiology and biochemical processes.

## 2. Materials and Methods

### 2.1. Plant Material and Growth Conditions

*A. thaliana* (Colombia) seeds and its *cipk3*/*9*/*23*/*26* quadruple mutant, which is very sensitive to moderate and high levels of external Mg^2+^ [6], were used in the present study. The seeds were sterilized with 10% (*v*/*v*) sodium hypochlorite for 14 min, and then washed three to four times with distilled water. Seeds were sown in Murashige and Skoog medium (pH 5.8). One week after germination, the seedlings were transferred to plastic trays with nutrient soil, which was a mixture of 70% turf, 20% perlite, and 10% vermiculite. WT and mutant seedlings were grown in an artificial illumination incubator with approximately 70% relative humidity, 200 μmol·m^−2^·s^−1^ photon flux density, and an 8 h light (21 °C): 16 h dark (19 °C) photoperiod. The seedlings were watered every three days. Four weeks after growth, the rosette leaves of two *A. thaliana* samples were collected and immediately preserved at −80 °C until analysis. Three biologically independent replicates of control and treatment samples were analyzed in the experiment.

### 2.2. Protein Extraction and Trypsin Digestion

One gram of each sample was placed in a pre-cooled mortar, and thoroughly ground into fine powder with liquid nitrogen. Four volumes of phenol extraction buffer (including 1% TritonX-100, 10 mM of dithiothreitol, 1% Protease Inhibitor Cocktail (Thermo Scientific™, Waltham, MA, USA), 50 μM PR-619, 3 μM TSA, 50 mM NAM, and 2 mM EDTA) were added to each group of leaf samples, followed by three ultrasonic treatments on ice using a high-intensity ultrasound processor (Scientz Biotechnology Co., Ningbo, China). A similar volume of tris-saturated phenol (pH 8.0) was added and centrifuged at 4 °C, 5000× *g*, for 10 min. The upper phenol phase was collected and five volumes of 0.1 M ammonium acetate in methanol were added, followed by incubation overnight. After centrifugation at 4 °C for 10 min, the protein precipitates were collected and washed once with ice-cold methanol, then three times with ice-cold acetone. The precipitates were subsequently redissolved in 8 M urea, and protein concentrations were determined using a BCA protein assay kit (Sigma-Aldrich, St. Louis, MO, USA).

For digestion, the protein solution was reduced with 5 mM of dithiothreitol for 30 min at 56 °C and alkylated with 11 mM of iodoacetamide for 15 min at room temperature in darkness. The protein sample was then diluted by adding 100 mM of TEAB to a urea concentration of less than 2 M. Finally, trypsin (Promega, Madison, WI, USA) was added at a 1:50 trypsin-to-protein mass ratio for the first digestion overnight and a 1:100 trypsin-to-protein mass ratio for a second 4 h digestion.

### 2.3. TMT Labeling and Reversed-Phase High-Performance Liquid Chromatography Fractionation

The peptides were desalted using the Strata X C18 solid-phase extraction column (Phenomenex, Torrance, CA, USA) after digestion with trypsin, then redissolved in 0.5 M triethylammonium bicarbonate. TMT labeling was performed using TMT mass tagging kits (ThermoFisher Scientific, Waltham, MA, USA) according to the manufacturer’s instructions: the thawed labeling reagents were dissolved in acetonitrile, and incubated for 2 h at 25 °C after mixing with the peptides. Finally, the peptide segment mixture was desalted and freeze-dried using vacuum centrifugation.

Subsequently, the peptide segment mixture obtained after enzymatic hydrolysis and desalination was separated using high-pH reversed-phase high-performance liquid chromatography on the Strata X C18 SPE column (Phenomenex, Torrance, CA, USA). The gradient was comprised of an increase from 6% to 23% of solvent B (0.1% formic acid in 98% acetonitrile) over 26 min, 23% to 35% in 8 min, and climbing to 80% in 3 min then holding at 80% for the last 3 min, all at a constant flow rate of 400 nL/min on an EASY-nLC 1000 UPLC system.

### 2.4. Modified Enrichment and LC-MS/MS Analyses

Phosphorylated peptide components were enriched using solid-phase metal ion affinity chromatography (IMAC). Briefly, peptides were dissolved in enrichment buffer (50% acetonitrile and 6% trifluoroacetic acid). The supernatant was aspirated into the IMAC material and incubated on a shaker. Subsequently, the peptide resin was washed three times with 50% acetonitrile (Pierce™, Waltham, MA, USA) in 6% trifluoroacetic acid and 30% acetonitrile in 0.1% trifluoroacetic acid to remove non-specifically adsorbed peptides, and 10% ammonia was added to elute the enriched phosphopeptides. Eluates containing phosphopeptides were collected, lyophilized, and desalted for LC-MS/MS analysis.

The enriched phosphopeptide components were analyzed using the EASY-nLC 1200 ultra-high-performance liquid chromatography system (Thermo Fisher Scientific, Waltham, MA, USA). Mobile phase A consisted of 0.1% (*v*/*v*) formic acid and 2% (*v*/*v*) acetonitrile, and mobile phase B consisted of 0.1% formic acid (Pierce^TM^, Waltham, MA, USA) and 90% acetonitrile. The liquid phase linear gradient was set as follows: 4–22% B for 38 min, 22–32% B for 14 min, 32–80% B for 4 min, and 80% B for 4 min. The flow rate was maintained at 450 nL/min.

The peptides were subjected to a NSI source followed by tandem mass spectrometry (MS/MS) in Q Exactive Plus (Thermo Scientific^TM^, Waltham, MA, USA) coupled online to the UPLC. The electrospray voltage applied was 2.0 kV. The m/z scan range was 350 to 1800 for a full scan, and intact peptides were detected in the Orbitrap at a resolution of 70,000. Peptides were then selected for MS/MS using the NCE setting as 28, and the fragments were detected in the Orbitrap at a resolution of 17,500. A data-dependent procedure alternated between one MS scan followed by 20 MS/MS scans with 15.0 s dynamic exclusion. Automatic gain control (AGC) was set at 5E4. The fixed first mass was set as 100 m/z.

### 2.5. Database Search and LC-MS/MS Data Analyses

The MS/MS spectra data were searched against the *A. thaliana* 3702 reference database (39,363 sequences) using Maxquant (v.1.5.2.8) with the following parameters: peptide tolerance of first search and main search were set to 20 and 10 ppm respectively, fragment ion mass tolerance was 0.02 Da, alkylation at cysteine was set as a fixed modification, oxidation at methionine, acetylation and deamidation at the N-terminal of protein, and phosphorylation at tyrosine, threonine, and serine were set as variable modifications. The enzymatic digestion model was trypsin/P, and two missing cleavages were allowed on trypsin. Using TMT 6-plex as the quantitative method, the false discovery rates of peptide-spectrum match identification and protein identification were set to 1%. To ensure high credibility of the results, the identified datasets were filtered based on a criterion of localization probability >0.75.

MoMo (http://meme-suite.org/tools/momo, accessed on 11 November 2019) was used to analyze motif characteristics of various phosphorylation modification sites. GO analyses were performed using the UniProt-GOA database (http://www.ebi.ac.uk/GOA/, accessed on 11 November 2019). If the UniProt-GOA database was not annotated, the GO function of the proteins was annotated using the InterPro domain database (http://www.ebi.ac.uk/interpro/, accessed on 11 November 2019) based on the protein sequence alignment method. The InterPro domain database was also used to analyze and identify functional descriptions of protein domains. Annotation analyses of protein pathways were conducted using the KEGG pathway database (http://www.genome.jp/kaas-bin/kaas_main; http://www.kegg.jp/kegg/mapper.html, accessed on 11 November 2019). The submitted proteins were annotated for subcellular localization using WoLF PSORT (http://www.genscript.com/psort/wolf_psort.html, accessed on 11 November 2019). In addition, the database numbers or protein sequences of the differentially modified proteins screened from various comparison groups were compared using the STRING 10.5 PPI network database (http://string-db.org/cgi/input.pl, accessed on 11 November 2019), and differential protein interaction relationships were extracted based on a confidence score of ≥0.7 (high confidence). Afterward, the “networkD3” package in R (https://cran.r-project.org/web/packages/networkD3/, accessed on 11 November 2019) was used to visualize protein interactions.

## 3. Results

### 3.1. Quantitative Phosphoproteomic Data Analysis

We applied IMAC enrichment and TMT labeling, coupled with liquid chromatography-tandem mass spectrometry (LC-MS/MS) approaches to identify cipk3/9/23/26-regulated phosphoproteomes in WT and *cipk3*/*9*/*23*/*26* quadruple mutant seedlings (Figure 1). Quality control evaluations of mass spectrometry data are illustrated in Figure 2. All points in the figure that represent peptides had mass errors within 10 ppm, which indicates that the data satisfied the requirements of high-precision characteristics of mass spectrometry (Figure 2A). Most of the peptides were distributed between seven and 20 amino acids, which was consistent with the general guidelines based on trypsin-induced enzymatic hydrolysis and higher energy collisional dissociation fragmentation (Figure 2B). Three statistical analysis methods, including principal component analysis, relative standard deviation, and Pearson’s correlation coefficient, were used to analyze the quantitative results of biological replicates. The analyses revealed that quantitative repeatability was high. Correlation between samples was satisfactory, and satisfied the expectation of the experimental design (Figure 2C).

### 3.2. Identification of Differential Phosphorylation Modification Sites and Proteins in the cipk3/9/23/26 Mutant and WT

In the present study, 12,506 phosphorylation sites were identified on 4537 proteins, of which 10,565 phosphorylation sites on 4258 proteins presented quantitative information (Appendix A). To ensure high credibility of the results, multiple comparisons were used to normalize the data. A difference in the degree of modification greater than 1.2 when the *p*-value was <0.05 was considered a significant upregulation, whereas a difference less than 1/1.2 was considered a significant downregulation. Finally, 1164 differential phosphorylation sites were identified among the 773 phosphoproteins, of which 607 sites on 376 phosphoproteins were upregulated and 557 sites on 397 phosphoproteins were downregulated (Figure 3A). Among the 1164 differential phosphorylation sites, 1011 sites were involved in the phosphorylation of serine (S) residues, 143 sites were involved in the phosphorylation of threonine (T) residues, and 10 sites were involved in the phosphorylation of tyrosine (Y) residues, accounting for 86.86%, 12.28%, and 0.86% of the total differential phosphorylation sites, respectively (Figure 3B, Appendix A). Notably, a comparison of the phosphorylation ratios of the three residue types in different plants and tissues revealed that the ratios calculated in the present study were very close to those reported previously [8,9,10], which suggests that the distribution of phosphorus among plant tissues was relatively conservative.

### 3.3. Motif Analysis of cipk3/9/23/26 Mutant Phosphoprotein Modification

Post-translational modification of proteins generally requires upstream kinases to recognize substrate-specific conserved amino acid motifs. Therefore, identifying and studying conserved motifs around phosphorylation sites is of great significance in the prediction of protein phosphorylation modifications and modification sites, as well as in the identification of cell signaling pathways. In the present study, the identified peptide sequences, which consisted of phosphorylation modification sites and their six upstream and downstream amino acids, were analyzed. A total of 8470 sequences were obtained, including 7688 (90.77%) phosphoserine-centered and 782 (9.23%) phosphothreonine-centered sequences (Appendix A). The Motif-X algorithm detected 59 over-represented motifs for phosphoserine and nine over-represented motifs for phosphothreonine (Table 1). The most prevalent motif for phosphoserine was [sP], which appeared 930 times and accounted for 12.17% of the identified motifs, followed by [Gs], [Rxxs], [sxE], and [sPR] motifs, which appeared 338, 314, 303, and 300 times, respectively. The two predominant motifs in phosphothreonine were [tP] and [PxtP]. Notably, no phosphotyrosine-centered sequences were obtained in the present study, which could be due to the low phosphotyrosine content that was not enriched in plant tissues.

### 3.4. Functional Classification of Differentially Phosphorylated Proteins

The identified differentially phosphorylated proteins (DPPs) were annotated and classified using the Gene Ontology (GO) approach based on three aspects: biological process, cellular component, and molecular function (Figure 4A). The principal classifications were based on ‘biological process’, ‘cellular process’, ‘metabolic process’, and ‘single-organism process’, while ‘behavior’, ‘biological adhesion’, and ‘locomotion’ were under-represented. In the category of ‘cellular component’, most of the differential phosphoproteins were associated with cells, organelles, and membranes, and only a few differential phosphoproteins occurred in the nucleoid. With regard to the ‘molecular function’ classification, differential phosphoproteins associated with catalytic activity and binding accounted for more than 80% of all identified phosphoproteins, and differential phosphoproteins associated with protein tagging and metallochaperone activity were not annotated.

Various organelles in plant tissues are often exposed to varying physical and chemical environments. Proteins can be specifically localized to various organelles based on their structural characteristics and functions. In the present study, WoLF PSORT was used to predict the subcellular structure and classify differentially phosphorylated proteins. The results revealed that differentially phosphorylated proteins were divided into 15 subcellular components, of which the nucleus (1740 proteins), chloroplast (1019 proteins), and cytoplasm (736 proteins) were the primary subcellular components (Figure 4B). Similarly, we used the UniProt-GOA database to perform a functional classification of differentially phosphorylated proteins, which was divided into 4 categories and 25 sub-categories: information storage and processing, cellular processes and signaling, metabolism, and poorly characterized. Detailed classification information is presented in Appendix A.

### 3.5. Differential Expression of Related Phosphoproteins between the cipk3/9/23/26 Mutant and WT

All the identified DPPs were analyzed statistically based on the distribution of GO terms to compare the differential expression of phosphoproteins and predict their biological functions. Among the upregulated proteins selected based on the GO terms, ‘cellular process’ (209 proteins) had the highest number of DPPs in ‘biological process’. The most dominant cellular component was ‘cell’ (267 proteins), while ‘binding’ (262 proteins) constituted the highest proportion in ‘molecular function’ (Figure 5A). Among the downregulated proteins, ‘cellular process’ (249 proteins), ‘cell’ (316 proteins), and ‘binding’ (270 proteins) were ranked among the top three major categories (Figure 5B). The classification results of GO enrichment revealed that ‘chloroplast relocation’, ‘establishment of plastid localization’, and ‘chloroplast localization’ were the most enriched GO terms in the biological process category of upregulated DPPs. Auxin efflux transmembrane transporter activity and Ca^2+^ binding were the primary molecular functions. ‘Cytoskeletal part’ was the most significantly enriched GO term in the cellular component category. Downregulated DPPs were mainly enriched in response to cytokinin, mRNA binding, chloroplast stroma, plastid stroma, and chloroplast.

### 3.6. Enrichment Analyses of DPPs

The Kyoto Encyclopedia of Genes and Genomes (KEGG) enrichment analyses revealed that the upregulated DPPs were closely associated with plant-pathogen interactions, ABC transporters, and carotenoid biosynthesis pathways. Downregulated DPPs were primarily closely associated with carbon fixation in photosynthetic organisms, glyoxylate and dicarboxylate metabolism, one-carbon pool by folate and glycine, and serine and threonine metabolism pathways (Figure 6A). The protein domain enrichment analyses revealed that the EF-hand domain, ABC transporter type 1, transmembrane domain, and ABC transporter-like proteins were significantly enriched in upregulated DPPs, while downregulated DPPs were closely associated with pyruvate phosphate dikinase, phosphoenolpyruvate/pyruvate-binding, glyceraldehyde 3-phosphate dehydrogenase, NAD(P)-binding domain, catalytic domains, and other aspects (Figure 6B).

### 3.7. Protein–Protein Interaction (PPI) Networks of DPPs

The samples were analyzed using the STRING 10.5 (v.10.5) protein interaction network database to explore interactions among DPPs, and the results were visualized using the networkD3 package in R. With the confidence score > 0.7 as the filtering criterion, a total of 209 DPPs were screened, including 77 upregulated and 132 downregulated proteins, and 1264 pairs of PPIs were obtained (Figure 7, Appendix A). The circles in Figure 7 represent the differentially modified proteins, shown by different colors (blue represents the downregulated protein and red represents the upregulated proteins). The circle size represents the number of differentially modified proteins and their interacting proteins. The larger the circle, the more proteins that interact with it, indicating that the protein is more important in the network. In order to clearly show the interaction between proteins, we screened the top 50 proteins with the closest interaction and mapped the protein interaction network.

## 4. Discussion

Protein phosphorylation is a key factor influencing the CBL-CIPK Ca^2+^ signaling network and Mg^2+^ homeostasis in *Arabidopsis*. Ca^2+^ signals can be detected by CBL2 and CBL3, which recruit and activate CBL-CIPK subsets. The CIPKs can interact with and phosphorylate Mg^2+^ transport systems to transport Mg^2+^ into vacuoles. Phosphorylated proteins with regard to membranes, transporters, and signal transductions were identified in the present study. The results could enhance our understanding of the *cipk3*/*9*/*23*/*26* mutant response to Mg^2+^ and signal transduction in *Arabidopsis*.

### 4.1. Response of the Cipk3/9/23/26 Mutant to Magnesium Toxicity

Mg^2+^ acts as a cofactor for enzymes and serves as a key chlorophyll metal ion in green tissues. High Mg^2+^ levels are toxic to the *cipk3*/*9*/*23*/*26* quadruple mutant, which not only results in severe growth retardation, but also causes chlorosis of leaf tips. Quantitative phosphoproteomics analyses revealed that the Mg-chelatase subunit, CHLI-1, was downregulated in the *cipk3*/*9*/*23*/*26* quadruple mutant, which could influence the loss of chlorophyll in leaf tips (Figure 1). Vacuolar (V-type) proton ATPase and vacuolar proton pyrophosphatase (V-PPase) take the responsibility for generating a proton gradient and membrane potential, which activate secondary transport across the vacuolar membrane [11]. In the *cipk3*/*9*/*23*/*26* quadruple mutant, the V-type proton ATPase subunit was downregulated, which in turn influenced the tonoplast membrane potential and consequent ion transport, resulting in high Mg^2+^ in the cytoplasm, which is toxic to the *cipk3*/*9*/*23*/*26* quadruple mutant.

### 4.2. Transport-Related Proteins

Three subfamilies (ABCB, ABCC, and ABCG) were detected in this study, which belong to members of the ABC transporter superfamily. Studies have uncovered that ABC transporters are not only mediated the transportation of hormones, lipids, metal ions, secondary metabolites, and exogenous substances in plants, but are also beneficial to plant–pathogen interactions, and the regulation of ion channels in plants [12]. Among them, ABCB transporters are mostly localized to the plasma membrane, and a few transporters are localized to the mitochondrial and chloroplast membranes. ABCB transporters can collaborate with auxin transporters to regulate polar auxin transport and are essential to plant growth and development. For example, ABCB1, ABCB4, ABCB14, ABCB19, and ABCB21 have been successively reported to be related to auxin transport [13,14,15,16,17]. ABCC transporters largely occur in vacuolar membranes, and their primary function is to transport a series of toxic substances to the vacuoles. Metal complexes with phytochelatins (PCs, known as class metallothioneins, which are induced in plants and are considered to detoxify metals by forming PC-metal complexes), and is transported into vacuoles by the ABCC transporter. The ABCC transporter is also associated with the regulation of ion channels and plant-mediated pathogenic responses [18]. The ABCC subfamily proteins have been associated with the transportation of a broad range of xenobiotics to the vacuoles. For example, AtABCC1 and AtABCC2 transporters are required for arsenic detoxification in *Arabidopsis,* and the C-type ATP-binding cassette transporter (OsABCC7) takes part in root-to-shoot translocation of arsenic in rice [19]. The ABCC subfamilies, which are vacuolar transporters, take part in lead detoxification in *Saccharomyces cerevisiae* [20]. ABCG transporters are functionally diverse. For example, ABCG transporter genes are expressed when plants are subjected to salt stress, tissue hypoxia, heavy metal stress, bacterial infection, and other external stresses [21,22]. In the present study, 10 ABC transporters that were upregulated at the phosphorylation level were identified, including the ABC transporter B family members 1, 6, 19, 21, and 27, ABC transporter C family members 1 and 14, and ABC transporter G family members 12, 22, and 36 (Table 2). In the ABCB subfamily, ABCB1, ABCB19, and ABCB21 expression levels, as well as PIN3, were upregulated, which suggests that the transporters could be involved in the regulation of auxin and ion transport. ABCG36 is a multifunctional protein in *Arabidopsis* that acts as a metal exporter [23]. The upregulated expression of ABCG36 in the present study suggests that it could be involved in maintaining Mg^2+^ balance in the *cipk3*/*9*/*23*/*26* mutant. In addition to ABC transporters, the expression levels of some transporters, such as vacuolar cation/proton exchanger 1 (CAX1), aquaporin (PIP2-6), cationic amino acid transporter 9 (CAT9), and potassium ion efflux antiporter 1 (KEA1), were upregulated to varying degrees (Table 2). Vacuolar cation/proton exchangers (CAXs) are universal types of ion transporters in plants. CAX transporters in *Arabidopsis* have been demonstrated to exhibit diverse roles in vacuolar sequestration of Ca^2+^ and other cations, presenting a vital mechanism for ion stress [24]. The *cipk3*/*9*/*23*/*26* mutant has a defective pathway for vacuolar Mg^2+^ sequestration, which increases toxicity in the cytoplasm under moderate and high Mg^2+^ levels. The general toxic effects of high Mg^2+^ in the *cipk3*/*9*/*23*/*26* mutant could impair Ca^2+^ homeostasis, which in turn inhibits Ca^2+^ uptake [6]. Mg^2+^ induces a specific Ca^2+^ signal, which is modulated by CAX. The signal is decoded by CBL2 and CBL3 sensors; however, CIPK3/9/23/26 does not interact with CBL2 and CBL3 in *cipk3*/*9*/*23*/*26*, and in turn, target proteins are not activated. Therefore, Mg^2+^ accumulates in the cytoplasm and is not sequestered into vacuoles, which leads to Mg^2+^ toxicity in plants.

Based on previous studies, variations in the phosphorylation states of the transporters in the *cipk3*/*9*/*23*/*26* mutant suggest that they could regulate ion concentrations to balance cell osmotic potential and facilitate the storage of Mg^2+^ in vacuoles, maintaining non-toxic Mg^2+^ levels in the cytoplasm.

### 4.3. Signal Transduction and Protein Kinases

Reversible protein phosphorylation, which is mediated by protein kinases and phosphatase, is one of the key post-translational modifications (PTMs) of proteins in signal transductions [25]. Mitogen-activated protein kinases (MAPKs) are serine/threonine protein kinases that are ubiquitous in eukaryotes. The resulting MAPK cascade pathways are the most widespread signaling pathways regulating plant growth, development, and stress response, which can transmit external stimuli into cells and enable cells to develop effective and timely defense responses. In the present study, we established that MKK1, MPK3, MPK6, MPK15, MPK16, and partial At1g80180 (MASS1) proteins, which were upregulated at the phosphorylation level, were associated with the MAPK cascade pathway, while MPK4 and partial At1g80180 proteins were downregulated at the phosphorylation level (Table 2). The MKK component is a member of the MAPKK gene family. MPK3, MPK4, and MPK6 are a family of MAPK protein kinases which are located downstream of the MAPK cascade signal. MPK3 and MPK6 are homologous proteins. Previous studies have revealed that the MEKK1-MKK1/MKK2-MPK4 cascade in *Arabidopsis* responds to cold and salt stress and negatively regulates immune responses mediated by SUMM2 [26,27,28,29]. In *Glycine max*, the GmMEKK1-GmMKK1/2/4/9-GmMPK3/6 cascade regulates cell death and participates in immune defense responses [30]. At1g80180 is a substrate protein of MPK3 and MPK6, and its overexpression can regulate the production and aggregation of stomata [31]. Specifically, the mechanisms of interaction between the MPK and MAPK signaling cascades and CBL-CIPK networks in the regulation of Mg^2+^ homeostasis require further study.

Ca^2+^ is a key second messenger in plant cell signal transduction. Ca^2+^ signals in the cell can be transmitted to downstream components by Ca^2+^ sensor proteins (CaM/CML/CBLs/CDPKs), exposing them to external stimuli, which can cause protein phosphorylation and the expression of related genes, thereby regulating plant growth and development, as well as immune and stress responses [32]. In the present study, CaM5, CML24, CPK3, and CPK9 were upregulated at the phosphorylation level, CPK10 and CPK13 were downregulated, while the expressions of CML35, CML41, CPK1, and CPK21 were upregulated or downregulated to varying degrees at the phosphorylation level (Table 2). Calmodulin (CaM) is a receptor protein that primarily mediates the regulation of enzymes, ion channels, and other proteins by Ca^2+^. Studies have revealed that CaM5 expression could be caused by fungal infections and mechanical damage [33,34]. CML is similar to CaM in structure and it is composed of EF-hand domains, and has the ability to bind to Ca^2+^. In *Arabidopsis*, AtCML24 can positively regulate polar growth of pollen tubes as a downstream receptor of Ca^2+^ signals, and take part in the regulation of plant root growth and development under mechanical stimulation [35,36]. AtCML41 can mediate the deposition of the corpus callosum by the Ca^2+^ signaling pathway to resist infection by pathogens [37]. CDPK is a class of serine/threonine protein kinases, which can interact with substrates without binding to other proteins due to having their own protein kinase domain. In the CDPK/Ca^2+^-mediated ABA signaling pathway, CPK3 and CPK6 positively regulate the ABA signaling pathway and are key factors influencing the regulation of guard cell anions, while CPK10 participates in the regulation of stomatal closure in *Arabidopsis* under drought conditions [38,39]. CPK21 exerts a negative regulatory effect in hyperosmotic stress [40]. In *Oryza sativa*, OsCPK9 can improve drought resistance in plants by promoting stomatal closure [41], and overexpression of the *OsCPK9* gene can enhance salt tolerance [42]. CDPK has a variety of substrates; among them, 14-3-3 proteins can either bind to phosphorylated CDPK or be phosphorylated by CDPK. In this study, two 14-3-3 proteins (GRF3 and GRF4) were upregulated at the phosphorylation level. Studies have revealed that peroxisome-bound CPK1 and CPK3 can phosphorylate 14-3-3 proteins at multiple sites, and that the 14-3-3 proteins exert direct regulatory effects on CDPK activity and stability in plant cells [43]. The results show that 14-3-3 proteins interact with CDPK and participate in stress response and signal transduction pathways, among others. The characterization of key signal transduction proteins suggests that phosphorylation enables cells to rapidly respond to Mg^2+^ sensitivity.

## 5. Conclusions

In the present study, 12,506 phosphorylation modification sites on 4537 proteins were identified in the *Arabidopsis*
*cipk3*/*9*/*23*/*26* mutant and WT using phosphorylation modification enrichment technology, and TMT labeling coupled with LC-MS/MS. We detected 773 DPPs (1164 differential phosphorylation sites), of which 376 were upregulated (607 sites) and 397 were downregulated (557 sites) based on a *p*-value < 0.05 and a change threshold of 1.2. We focused on the global variations in phosphorylated proteins involved in transportation, signal transduction, and protein kinases in various cellular pathways and processes. The results of the study revealed that the abundance of phosphopeptides of certain ABC transporters, CAX1, PIP2-6, CAT9, and KEA1, were upregulated to varying degrees, in addition to the upregulation of several protein kinases, such as MAPKs and CDPKs. The results are a fundamental resource, which provide insights into how the *cipk3*/*9*/*23*/*26* mutant plants respond to Mg^2+^ sensitivity and adapt to physical and chemical changes, thereby maintaining Mg^2+^ homeostasis, and a balance in the physiological and biochemical processes of the cellular components. The research of PTM events would elucidate a major checkpoint for cellular signaling in plant responses to Mg^2+^ sensitivity.

## Figures and Tables

**Figure 1 genes-12-01759-f001:**
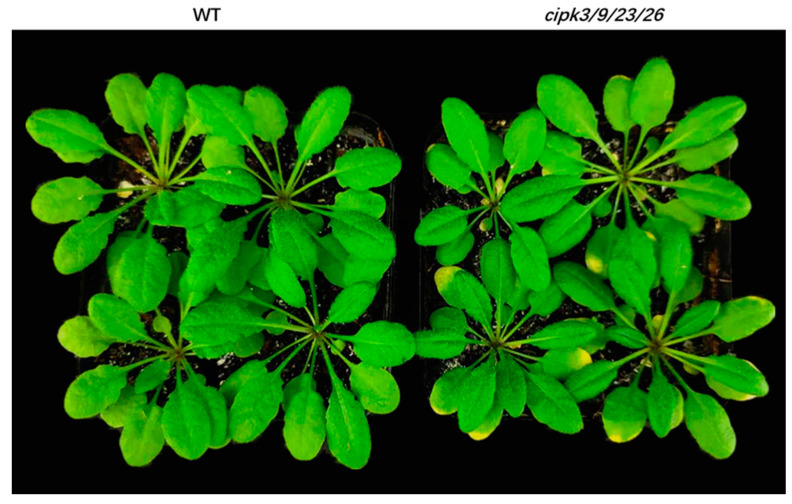
Wild type *Arabidopsis thaliana* (Colombia wild) and its *cipk3*/*9*/*23*/*26* mutant.

**Figure 2 genes-12-01759-f002:**
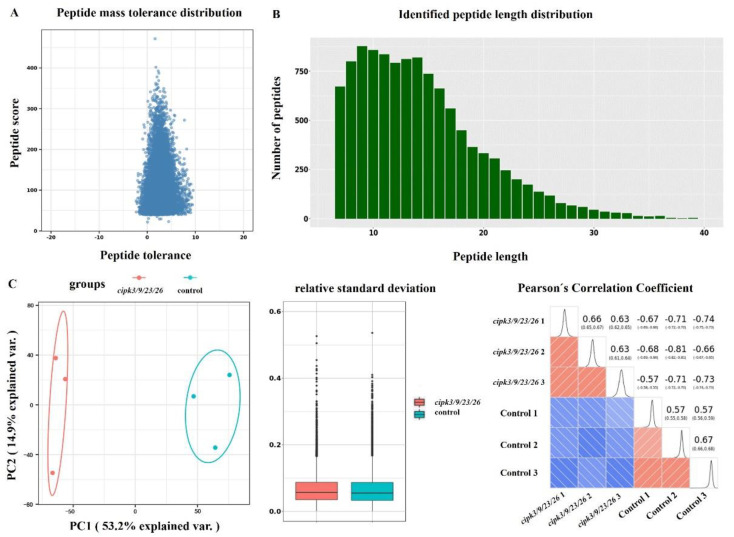
Quality control (QC) evaluation of mass spectrometry (MS) data. (**A**) Mass error distribution of phosphorylated peptides identified by mass spectrometry. (**B**) Length distribution of all phosphorylated peptides identified by mass spectrometry. (**C**) Modified quantitative principal component analyses, relative standard deviation, and Pearson’s correlation coefficient among repeated samples.

**Figure 3 genes-12-01759-f003:**
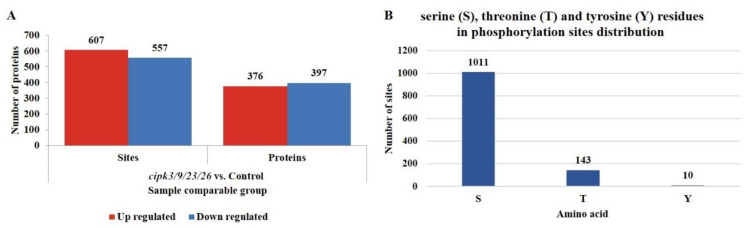
Identification of phosphorylation modification sites and proteins. (**A**) The number distributions of differentially phosphorylated proteins and phosphorylation sites in various comparison groups. (**B**) Serine (S), threonine (T), and tyrosine (Y) residues in phosphorylation site distributions.

**Figure 4 genes-12-01759-f004:**
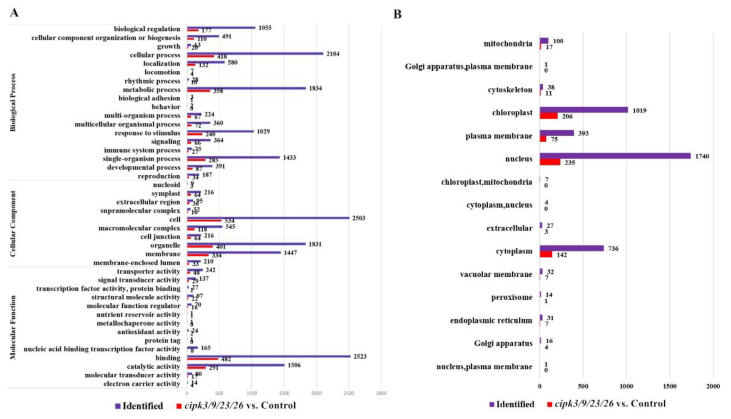
Classification of all identified phosphorylated proteins and differentially phosphorylated proteins (DPPs). (**A**) Gene Ontology (GO) analyses of all identified phosphorylated proteins and DPPs. All proteins were classified using GO terms based on three categories: molecular function, biological process, and cellular component. (**B**) Subcellular structure classification of phosphorylated protein and DPPs.

**Figure 5 genes-12-01759-f005:**
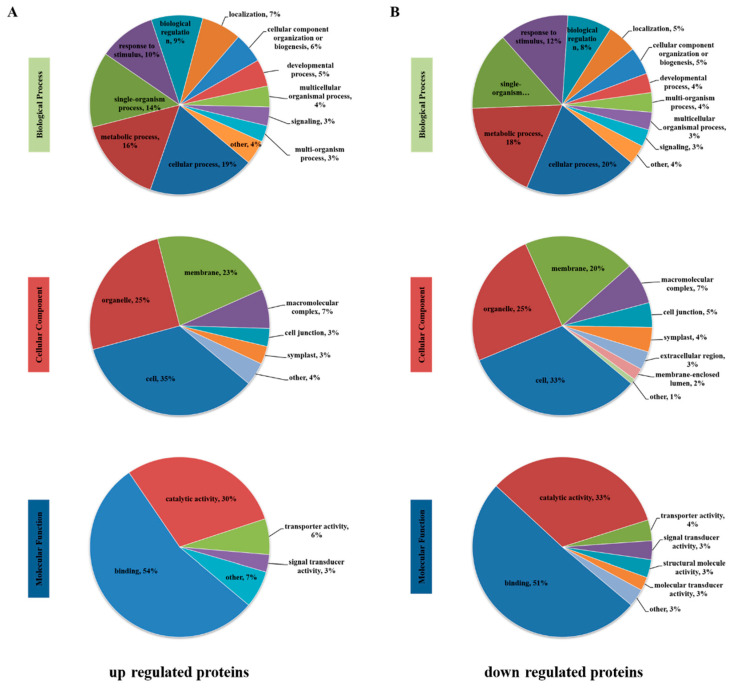
GO annotation analyses of DPPs. (**A**) Distribution of upregulated proteins identified using the GO annotation tool. (**B**) Distribution of downregulated proteins identified using the GO annotation tool. Different color blocks represent different terms, including cellular component, molecular function, and biological process.

**Figure 6 genes-12-01759-f006:**
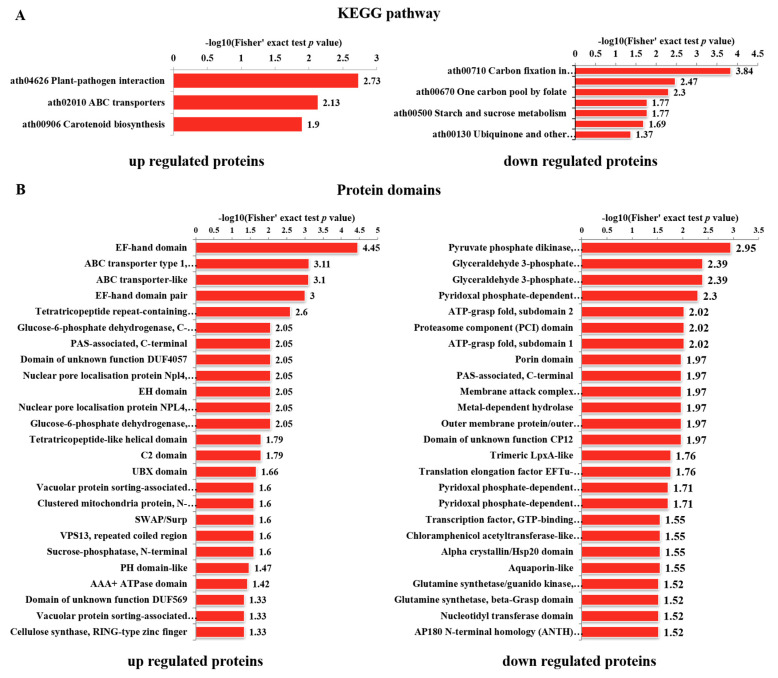
Kyoto Encyclopedia of Genes and Genomes (KEGG) and domain enrichment analyses of DPPs in *Arabidopsis thaliana*. (**A**) Significantly enriched KEGG pathways associated with the DPPs. (**B**) Significantly enriched protein domains associated with the DPPs.

**Figure 7 genes-12-01759-f007:**
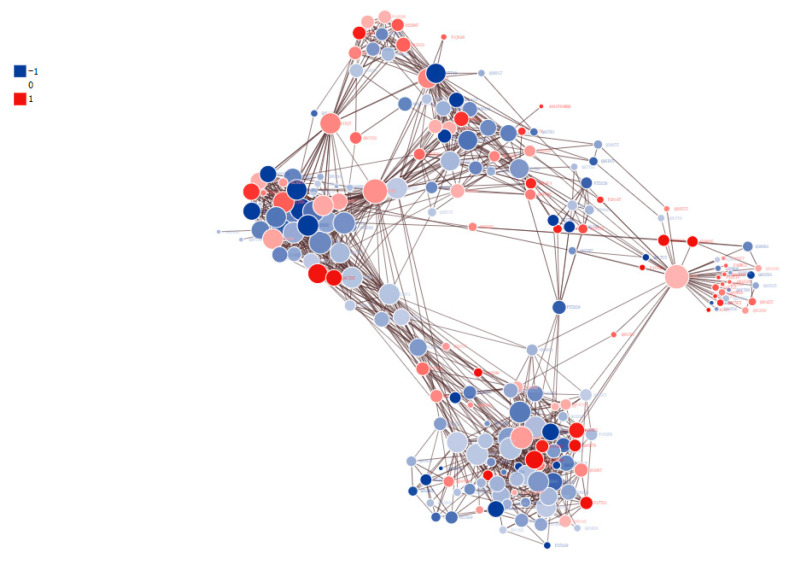
Protein–protein interaction network of DPPs. Different circles in the figure represent different DPPs, in which red is the upregulated protein and blue is the downregulated protein. The larger the circle, the greater the number of proteins that interact with it, indicating that the protein is more important in the network.

**Table 1 genes-12-01759-t001:** Over-represented motifs detected using the Motif-X algorithm.

Motif	Motif Score	Foreground		Background		Fold Increase
		Matches	Size	Matches	Size	
xxxxPx_S_PxRxxx	39.87	61	8351	255	952,659	27.3
xxxxPx_S_PRxxxx	40.29	53	8290	219	952,404	27.8
xxxxPx_S_PKxxxx	39.63	41	8237	169	952,185	28.0
xxxxxx_S_PRxxxx	32.00	300	8196	2586	952,016	13.5
xxxxPx_S_Pxxxxx	32.00	281	7896	3314	949,430	10.2
xxxxxx_S_PxRxxx	32.00	246	7615	2235	946,116	13.7
xxxRxx_S_PxPxxx	38.69	47	7369	229	943,881	26.3
xxxxxx_S_PxxxRx	32.00	199	7322	2288	943,652	11.2
xxxxxx_S_PKxxxx	29.51	163	7123	2114	941,364	10.2
xxxRSx_S_xPxxxx	38.04	45	6960	388	939,250	15.7
xxxxxx_S_PxxxxR	30.02	149	6915	1948	938,862	10.4
xLxRxx_S_xxxxxx	32.00	292	6766	5590	936,914	7.2
xxxxxR_S_Pxxxxx	27.82	112	6474	1486	931,324	10.8
xxxxxG_S_Pxxxxx	27.98	124	6362	1797	929,838	10.1
xxxRSx_S_xxxxxx	32.00	234	6238	5119	928,041	6.8
xxxxxx_S_PxxRxx	25.38	93	6004	1423	922,922	10.0
xxxxxx_S_Pxxxxx	16.00	930	5911	29,886	921,499	4.9
xxxRxx_S_Fxxxxx	32.00	116	4981	1886	891,613	11.0
xxxRxx_S_xDxxxx	28.35	85	4865	1907	889,727	8.2
xxxxxx_S_DDExxx	38.28	46	4780	478	887,820	17.9
xxxxxx_S_DGExxx	39.07	28	4734	235	887,342	22.3
xMxRxx_S_xxxxxx	25.45	48	4706	986	887,107	9.2
xxxxxx_S_DxExxx	32.00	112	4658	3395	886,121	6.3
xxxRxx_S_xPxxxx	24.23	63	4546	1741	882,726	7.0
xxxxxG_S_Gxxxxx	25.21	99	4483	5462	880,985	3.6
xLxKSx_S_xxxxxx	38.37	29	4384	497	875,523	11.7
xxxxxD_S_DxDxxx	39.45	37	4355	422	875,026	17.6
xxxRxx_S_xExxxx	23.85	65	4318	2050	874,604	6.4
xxxKxx_S_Fxxxxx	32.00	85	4253	2802	872,554	6.2
xxxRxx_S_xGxxxx	32.00	66	4168	2312	869,752	6.0
xxxxxx_S_ExExxx	32.00	105	4102	5008	867,440	4.4
xxxxxx_S_DxGxxx	24.52	55	3997	2966	862,432	4.0
xxxxxx_S_DxDxxx	32.00	86	3942	3168	859,466	5.9
xxxxxG_S_Fxxxxx	25.25	52	3856	2487	856,298	4.6
xxxxRx_S_xDxxxx	26.26	54	3804	2257	853,811	5.4
xxxxxx_S_ExGxxx	24.29	53	3750	3143	851,554	3.8
xLxKxx_S_xxxxxx	24.76	71	3697	4574	848,411	3.6
xxxxSx_S_Fxxxxx	29.03	68	3626	4094	843,837	3.9
xxxxxG_S_xxxxxx	16.00	338	3558	47,187	839,743	1.7
xxxRxx_S_xxxxxx	16.00	314	3220	25,377	792,556	3.0
xxxxSR_S_xxxxxx	24.33	54	2906	4112	767,179	3.5
xxxxxx_S_xExxxx	16.00	303	2852	43,739	763,067	1.9
xxxxxx_S_xGxxxx	16.00	285	2549	48,192	719,328	1.7
xxxxxx_S_xPxSPx	39.01	29	2264	388	671,136	22.2
xxxxxx_S_xDxxxx	16.00	261	2235	37,147	670,748	2.1
xxxxxx_S_GPLxxx	39.06	24	1974	245	633,601	31.4
xxRxxx_S_xPxxxx	23.73	34	1950	2060	633,356	5.4
xxxxRx_S_xSxxxx	29.92	74	1916	4763	631,296	5.1
xxxxxx_S_FRxxxx	27.42	33	1842	1453	626,533	7.7
RxxSxx_S_xxxxxx	20.27	49	1809	4205	625,080	4.0
xxxxxD_S_xxxxxx	12.49	167	1760	32,542	620,875	1.8
xxxxxx_S_xPxxxx	13.05	172	1593	34,988	588,333	1.8
xxxKxx_S_xxxxxx	10.30	139	1421	30,182	553,345	1.8
xxxxxx_S_xRxxxx	12.48	149	1282	32,477	523,163	1.9
xxxxxx_S_Fxxxxx	9.38	99	1133	21,946	490,686	2.0
xxxxSx_S_xNxxxx	15.13	31	1034	3443	468,740	4.1
xxxxxx_S_Dxxxxx	7.27	82	1003	20,145	465,297	1.9
xxxxxx_S_xxGxxx	7.96	98	921	25,916	445,152	1.8
xxxxxx_S_xKxxxx	8.29	115	823	33,536	419,236	1.7
xxxxPx_T_Pxxxxx	32.00	143	1057	2335	529,615	30.7
xxxxxx_T_PxRxxx	27.88	63	914	1278	527,280	28.4
xxxxxx_T_PRxxxx	24.65	48	851	1115	526,002	26.6
xxxxxx_T_PTxxxx	24.66	54	803	1470	524,887	24.0
xxxxxx_T_PKxxxx	23.80	47	749	1406	523,417	23.4
xxxxxx_T_Pxxxxx	16.00	239	702	19,445	522,011	9.1
xxxRxx_T_xxxxxx	16.00	83	463	25,391	502,566	3.5
xxxxxx_T_xExxxx	10.57	63	380	31,767	477,175	2.5
xxxxxx_T_Dxxxxx	6.44	38	317	21,640	445,408	2.5

Motif: simplified form of motifs; Motif Score: log odds score of the motif matrix, higher scores are better matches; Foreground Matches: indicates the number of peptides containing a given motif in the identified modification site peptides after removing all the peptides containing the previously extracted motifs; Background Matches: indicates the number of peptides with a given motif in the same length of peptides composed of amino acids that can be modified in the input database full protein sequence, after removing all the peptides containing the previously extracted motifs; Foreground Size: indicates the number of remaining peptides in the identified modified site peptides after removing all peptides containing the previously extracted motifs; Background Size: indicates the number of peptides remaining in the same-length peptides composed of amino acids that can be modified in the input database full protein sequence after removing all the peptides containing the previously extracted motifs; Fold Increase: to assess the enrichment level of the extracted motifs.

**Table 2 genes-12-01759-t002:** Signaling and transport-related proteins differentially expressed in *cipk3*/*9*/*23*/*26* leaves at the phosphorylation level.

Protein Accession	Position	Ratio	Protein Description	Modified Sequence
Q9ZR72	1014	1.49	ABC transporter B family member 1	KTEIEPDDPDT(0.203)T(0.797)PVPDR
Q9ZR72	642	1.386	ABC transporter B family member 1	NS(0.016)S(0.979)Y(0.005)GRS(0.965)PY(0.005)S(0.03)R
Q9LJX0	624	1.262	ABC transporter B family member 19	T(0.157)RS(0.831)T(0.094)RLS(0.917)HS(0.32)LS(0.672)T(0.008)K
Q9LJX0	620	1.405	ABC transporter B family member 19	T(0.157)RS(0.831)T(0.094)RLS(0.917)HS(0.32)LS(0.672)T(0.008)K
Q9LJX0	611	1.335	ABC transporter B family member 19	DFS(0.999)NPS(0.001)TR
Q9M1Q9	660	1.554	ABC transporter B family member 21	LSMES(1)MKR
Q0WML0	639	1.252	ABC transporter B family member 27	QLQS(0.009)S(0.086)S(0.889)S(0.016)VTTL
Q9C8G9	1485	1.778	ABC transporter C family member 1	S(0.007)IT(0.993)LENKR
Q9LZJ5	897	1.403	ABC transporter C family member 14	SIS(1)IES(1)PRQPKS(1)PK
Q9LZJ5	894	1.615	ABC transporter C family member 14	SIS(1)IES(1)PRQPKS(1)PK
Q9LZJ5	903	1.345	ABC transporter C family member 14	SIS(1)IES(1)PRQPKS(1)PK
Q9C8K2	667	1.237	ABC transporter G family member 12	KVPS(0.003)LS(0.162)S(0.162)LS(0.68)S(0.994)RR
Q9C8K2	666	1.289	ABC transporter G family member 12	KVPSLS(0.003)S(0.009)LS(0.9)S(0.088)RR
Q9C8K2	661	1.245	ABC transporter G family member 12	KVPS(0.999)LS(0.002)S(0.009)LS(0.829)S(0.162)R
Q93YS4	71	1.29	ABC transporter G family member 22	LMGMS(0.996)PGRS(0.175)S(0.806)GAGT(0.022)HIR
Q93YS4	66	1.428	ABC transporter G family member 22	RLMGMS(1)PGR
Q9XIE2	40	1.266	ABC transporter G family member 36	NIEDIFSS(0.003)GS(0.997)R
A0A1P8AZ84	586	2.017	p-glycoprotein 6	QKS(0.888)NGS(0.116)DPES(0.988)PIS(0.007)PLLISDPQNER
A0A1P8AZ84	610	1.419	p-glycoprotein 6	S(0.002)HS(0.997)QT(0.001)FSRPLGHSDDTSASVK
A0A1P8AZ84	593	1.548	p-glycoprotein 6	QKS(0.888)NGS(0.116)DPES(0.988)PIS(0.007)PLLISDPQNER
Q9S7Z8	235	1.326	Auxin efflux carrier component 3	PSNLTGAEIYS(1)LS(0.003)T(0.201)T(0.797)PR
F4IS06	38	1.674	Vacuolar cation/proton exchanger	TAHNMS(0.993)S(0.119)S(0.539)S(0.348)LRK
Q9ZV07	282	1.214	Probable aquaporin PIP2-6	S(1)QLHELHA
A0A1P8AU30	30	1.27	Cationic amino acid transporter 9	S(0.012)KS(0.986)LPPPS(0.001)S(0.001)QT(0.001)AVR
Q9ZTZ7	120	1.733	K(+) efflux antiporter 1, chloroplastic	IGES(0.002)S(0.013)ES(0.864)S(0.121)DETEATDLK
Q8W4J2	527	1.374	Mitogen-activated protein kinase 16	T(0.237)QPCKS(0.763)NRGDEDCATAAEGPSR
Q94A06	27	1.235	Mitogen-activated protein kinase kinase 1	FLT(0.096)QS(0.9)GT(0.005)FKDGDLR
Q9C9U4	530	1.742	Mitogen-activated protein kinase 15	ASQQAEGTENGGGGGYS(1)AR
Q39026	221	2.126	Mitogen-activated protein kinase 6	VTSESDFMT(1)EY(1)VVTR
Q39026	223	2.106	Mitogen-activated protein kinase 6	VTSESDFMT(1)EY(1)VVTR
Q39023	198	2.156	Mitogen-activated protein kinase 3	PTSENDFMT(1)EY(1)VVTR
Q39023	196	2.309	Mitogen-activated protein kinase 3	PTSENDFMT(1)EY(1)VVTR
Q39024	195	0.637	Mitogen-activated protein kinase 4	T(0.036)KS(0.908)ET(0.056)DFMTEYVVTR
Q9SSC1	98	1.208	MAPK kinase substrate protein At1g80180	VS(1)PAVDPPS(1)PR
Q9SSC1	16	1.429	MAPK kinase substrate protein At1g80180	RQGS(1)S(1)GIVWDDR
Q9SSC1	17	1.33	MAPK kinase substrate protein At1g80180	RQGS(1)S(1)GIVWDDR
Q9SSC1	82	0.464	MAPK kinase substrate protein At1g80180	S(0.013)RS(0.987)NGGGAIR
F4IVN6	80	1.251	Calmodulin 5	MKDT(0.998)DS(0.002)EEELK
F4IVN6	82	1.21	Calmodulin 5	MKDT(0.017)DS(0.983)EEELK
P30188	27	1.812	Probable calcium-binding protein CML35	AS(0.008)VS(0.183)RS(0.803)EPS(0.501)S(0.501)FS(0.002)S(0.002)NASSSSSDGSYGNLK
P30188	11	0.765	Probable calcium-binding protein CML35	LAASLNRLS(1)PK
P30188	44	0.208	Probable calcium-binding protein CML35	SEPSSFSSNASSSSSDGS(1)YGNLK
P25070	46	1.215	Calcium-binding protein CML24	ALS(0.937)PT(0.059)AS(0.004)PEETVTMMK
Q8L3R2	159	0.593	Probable calcium-binding protein CML41	GSGCIT(1)PK
Q8L3R2	26	1.321	Probable calcium-binding protein CML41	LNLS(1)FQNR
Q8L3R2	47	0.566	Probable calcium-binding protein CML41	SNSS(0.001)S(0.003)T(0.017)LNS(0.98)PRS(1)NSDDNNNIK
Q38868	69	1.208	Calcium-dependent protein kinase 9	AAAAAPGLS(1)PK
Q38868	51	1.627	Calcium-dependent protein kinase 9	TTQQPEKPGS(0.997)VNS(0.003)QPPPWR
Q38868	78	1.889	Calcium-dependent protein kinase 9	S(0.001)NS(0.999)ILENAFEDVK
Q9ZSA2	244	0.432	Calcium-dependent protein kinase 21	DIVGS(1)AYYVAPEVLR
Q9ZSA2	53	1.322	Calcium-dependent protein kinase 21	PMTQPIHQQIS(0.964)T(0.036)PSSNPVSVR
Q9ZSA2	414	1.449	Calcium-dependent protein kinase 21	LGS(0.997)RLS(0.003)ETEVK
Q9ZSA2	417	0.744	Calcium-dependent protein kinase 21	LS(0.971)ET(0.029)EVK
Q9M9V8	40	0.394	Calcium-dependent protein kinase 10	LNPFAGDFT(0.002)RS(0.998)PAPIR
Q8W4I7	18	0.712	Calcium-dependent protein kinase 13	EDVKS(1)NY(0.001)S(0.999)GHDHAR
Q8W4I7	21	0.454	Calcium-dependent protein kinase 13	SNYS(1)GHDHAR
Q8W4I7	43	0.511	Calcium-dependent protein kinase 13	VLS(1)DVPKENIEDR
Q42479	18	1.485	Calcium-dependent protein kinase 3	SSDPPPSS(0.005)S(0.768)S(0.175)S(0.041)S(0.009)S(0.002)GNVVHHVKPAGER
Q06850	130	0.778	Calcium-dependent protein kinase 1	RVS(0.076)S(0.924)AGLR
Q06850	64	1.878	Calcium-dependent protein kinase 1	LSDEVQNKPPEQVT(0.997)MPKPGT(0.003)DVETK
Q06850	129	0.549	Calcium-dependent protein kinase 1	RVS(0.994)S(0.006)AGLR
P46077	248	1.31	14-3-3-like protein GF14 phi	DNLTLWTSDMQDES(1)PEEIKEAAAPKPAEEQK
P42644	238	1.575	14-3-3-like protein GF14 psi	DNLTLWTSDMT(1)DEAGDEIK

## Data Availability

No new data were created or analyzed in this study. Data sharing is not applicable to this article.

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
