# Peer review of "Quantitative Phosphoproteomics of cipk3/9/23/26 Mutant and Wild Type in Arabidopsis thaliana"

_genes, 2021, doi:10.3390/genes12111759_

Round 1
Reviewer 1 Report
In this article, the authors used phosphoproteomics methods to analyze the Mg2+ responses in the cipk mutants. 773 differentially phosphorylated proteins were identified. The methods are profound, the use of TMT labeling and multiplex are proficient, and the analysis of proteomics data is reasonable. However, this research lacks experimental data to validate the proteomics results, and the discussion failed to fully discuss the results. The detailed comments are as followed:
- The key thing missing in this research is the validation data. For example, western blot showing a Mg2+ marker protein phosphorylated/dephosphorylated as expected? Or targeted-MS results?
- The authors made a lot of discussion on key regulators in the Mg2+ regulation pathways, but not much about the presented data. For example, the authors listed a lot of motifs identified by the software, and a group of protein regulation network was shown. But I didn't find more discussion on these data.
- For Figure 7, is there any grouping of these DPPs? Or there’re words but just too blur too read at this point?
- The article need more proofreading. For example, the reference style is weird, please revise it to be consistent and as required by the journal. And some details of drug resources were missing, for example, in section 2.2. Also a lot of blanks were missing between words, making it confusing to read.
Author Response
Dear reviewer,
Thank you for your suggestions and comments on our manuscript entitled “Quantitative phosphoproteomics of cipk3/9/23/26 mutant and wild type in Arabidopsis thaliana”. Those comments are very helpful for revising and improving our paper.
Replies to the reviewers’ comments:
- The key thing missing in this research is the validation data. For example, western blot showing a Mg2+ marker protein phosphorylated/dephosphorylated as expected? Or targeted-MS results?
Response to: It is really true as Reviewer suggested that our manuscript did not provide targeted-MS results. In the next study, we will use PRM to verify the Mg2+marker proteins, and study the protein interaction.
- The authors made a lot of discussion on key regulators in the Mg2+ regulation pathways, but not much about the presented data. For example, the authors listed a lot of motifs identified by the software, and a group of protein regulation network was shown. But I didn't find more discussion on these data.
Response to: It is really true as Reviewer’s comments. Next work, we would study the key regulators’ interaction to verify the proteins and motifs identified by the software.
- For Figure 7, is there any grouping of these DPPs? Or there’re words but just too blur too read at this point?
Response to: It is really true as Reviewer’s comment that words for Figure 1 are too blur too read. In the revised manuscript, some words were added to explain Figure 7.
- The article need more proofreading. For example, the reference style is weird, please revise it to be consistent and as required by the journal. And some details of drug resources were missing, for example, in section 2.2. Also a lot of blanks were missing between words, making it confusing to read.
Response to: We are very sorry for the reference style and missing spaces. They would be revised for Genes style. Details of drug resources were added.
Once again, thank you very much for your constructive comments and suggestions which would help us both in English and in depth to improve the quality of the paper.
Best regards
Yan Zhen
Reviewer 2 Report
Results presented by Authors are valuable, interesting and novel to the field of magnesium ion homeostasis regulation in plants.
The main weakness of this manuscript is relatively large number of typographical errors- common is a lack of space between words, problems with citation form- particularly in Discussion section. Some sentences has a typographical errors that destroy their sense: for example (page 7) The ABCG transporter is multifunctional protein in Arabidopsis that acts as heavy mental (should be metal) transporter.
Latin names of plants should be in italics.
All these relatively small mistakes may be relatively easy removed and corrected by Authors in the entire text.
Moreover section 2.3 lack information of flow rate, and duration/time of the gradient stages.
Author Response
Dear reviewer,
Thank you for your suggestions and comments on our manuscript entitled “Quantitative phosphoproteomics of cipk3/9/23/26 mutant and wild type in Arabidopsis thaliana”. Those comments are very helpful for revising and improving our paper.
Replies to the reviewers’ comments:
- The main weakness of this manuscript is relatively large number of typographical errors- common is a lack of space between words, problems with citation form- particularly in Discussion section. Some sentences have typographical errors that destroy their sense: for example (page 7) The ABCG transporter is multifunctional protein in Arabidopsis that acts as heavy mental (should be metal) transporter.
Response to: We are very sorry for missing spaces and typographical errors. They were revised in this manuscript.
- Latin names of plants should be in italics.
Response to: Yes, we revised the Latin names of plants.
- All these relatively small mistakes may be relatively easy removed and corrected by Authors in the entire text.
Response to: We are very sorry for these small mistakes. They were revised in this manuscript.
- Moreover section 2.3 lack information of flow rate, and duration/time of the gradient stages.
Response to: The flow rate and duration/time of the gradient stages were added in this manuscript.
Once again, thank you very much for your constructive comments and suggestions which would help us both in English and in depth to improve the quality of the paper.
Best regards
Yan Zhen